# Circulating microRNAs Suggest Networks Associated with Biological Functions in Aggressive Refractory Type 2 Celiac Disease

**DOI:** 10.3390/biomedicines10061408

**Published:** 2022-06-14

**Authors:** Nicoletta Bianchi, Luisa Doneda, Luca Elli, Cristian Taccioli, Valentina Vaira, Alice Scricciolo, Vincenza Lombardo, Anna Terrazzan, Patrizia Colapietro, Leonardo Terranova, Carlo Bergamini, Maurizio Vecchi, Lucia Scaramella, Nicoletta Nandi, Leda Roncoroni

**Affiliations:** 1Department of Translational Medicine, University of Ferrara, Street L. Borsari 46, 44121 Ferrara, Italy; nicoletta.bianchi@unife.it (N.B.); anna.terrazzan@unife.it (A.T.); 2Department of Biomedical, Surgical and Dental Sciences, University of Milan, Street Pascal 36, 20133 Milan, Italy; luisa.doneda@unimi.it (L.D.); leda.roncoroni@unimi.it (L.R.); 3Center for Prevention and Diagnosis of Celiac Disease, Gastroenterology and Endoscopy Unit, Fondazione IRCCS Ca’ Granda, Ospedale Maggiore Policlinico, 20122 Milan, Italy; alice.scricciolo@policlinico.mi.it (A.S.); vincenza.lombardo@policlinico.mi.it (V.L.); maurizio.vecchi@unimi.it (M.V.); lucia.scaramella@policlinico.mi.it (L.S.); nicoletta.nandi@unimi.it (N.N.); 4Department of Animal Medicine, Production and Health, University of Padova, 35020 Legnaro, Italy; cristian.taccioli@unipd.it; 5Division of Pathology, Fondazione IRCCS Ca’ Granda, Ospedale Maggiore Policlinico, Street F. Sforza 35, 20122 Milan, Italy; valentina.vaira@unimi.it; 6Department of Pathophysiology and Transplantation, University of Milan, Street F. Sforza 35, 20122 Milan, Italy; patrizia.colapietro@unimi.it; 7Respiratory Unit and Cystic Fibrosis Adult Center, Internal Medicine Department, Fondazione IRCCS Ca’ Granda, Ospedale Maggiore Policlinico, Street F. Sforza 35, 20122 Milan, Italy; leonardo.terranova@policlinico.mi.it; 8Department of Neuroscience and Rehabilitation, University of Ferrara, Street L. Borsari 46, 44121 Ferrara, Italy; carlo.bergamini@unife.it

**Keywords:** celiac disease, refractory celiac disease, gluten-free diet, microRNAs

## Abstract

Despite following a gluten-free diet, which is currently the only effective therapy for celiac disease, about 5% of patients can develop serious complications, which in the case of refractory type 2 could evolve towards intestinal lymphoma. In this study, we have identified a set of 15 microRNAs in serum discriminating between the two types of refractory disease. Upregulated miR-770-5p, miR-181b-2-3p, miR-1193, and miR-1226-3p could be useful for the better stratification of patients and the monitoring of disease development, while miR-490-3p was found to be dysregulated in patients with refractory type 1. Finally, by using bioinformatic tools applied to the analysis of the targets of dysregulated microRNAs, we have completed a more precise assessment of their functions. These mainly include the pathway of response to Transforming Growth Factor β cell–cell signaling by Wnt; epigenetic regulation, especially novel networks associated with transcriptional and post-transcriptional alterations; and the well-known inflammatory profiles.

## 1. Introduction

Celiac disease (CD) is an autoimmune enteropathy that occurs in genetically susceptible individuals characterized by HLA-DQ2 or HLA-DQ8 haplotypes after the ingestion of gluten. The disease proceeds through an inflammatory process in the bowel mucosa, leading to villous atrophy, crypt hyperplasia, and the infiltration of lymphocytes and plasma cells in the lamina propria. The antigenic trigger is gluten, a protein complex present in grains of wheat, barley, and rye and consisting of two major protein types, gliadins and glutenins. A lifelong gluten-free diet (GFD) is the only currently available treatment for CD [1]. Circulating microRNAs (miRNAs), which are short non-coding RNAs (19–24 nucleotides) implicated in post-transcriptional gene regulation, may be candidate biomarkers for the early detection of CD, could be useful to monitor GFD adherence, and are also associated with disease-stage-specificity [2]. Several studies have discussed the role of miRNAs in autoimmune disorders. Currently, the dysregulation of some miRNAs, such as miRNA-146a, miRNA-155, miRNA-21, and miRNA-125b, is detected mainly in the intestinal mucosa but has also been found in the peripheral blood cells and plasma of CD patients [3]. However, these miRNAs have also been demonstrated in several autoimmune diseases (e.g., in Alzheimer’s, psoriasis, and rheumatoid arthritis), but they do not seem to be specific as they are associated with an inflammatory state rather than a particular diagnosis [3]. Buoli Comani et al. described two types of miRNAs, miR-21-3p and miR-21-5p, which are strongly upregulated in the small intestine of CD patients, as well as in the circulation [4]. Moreover, Bascuñán et al. demonstrated that miR-21 is not useful for monitoring the response to a GFD because it did not normalize after the patient had started therapy [3], while six downregulated miRNAs (e.g., miR-150-3p and miR-150-5p) showed an increasing trend upon GFD. These data suggest that miRNAs are potential markers for CD and could be effective for supervising the response to a GFD. In addition, 53 circulating miRNAs were identified as possible early novel biomarkers, eight of which (miR-21-3p, miR-374a-5p, miR-144-3p, miR-500a-3p, miR-486-3p, let-7d-3p, let-7e-5p, and miR-3605-3p) have been detected at levels above the upper limits of normal and more than 2 years before the increase in transglutaminase antibodies (TGA) [2]. Another relevant aspect of miRNAs in CD patients concerns their expression in association with specific clinical features and in the pathogenesis of intestinal barrier dysfunction [1,5]. Approximately 5% of CD patients may develop serious and life-compromising complications such as refractory CD (RCD) and enteropathy-associated T-cell lymphoma (EATL) [6], characterized by villous atrophy and persistent malabsorption symptoms, even after a GFD is followed for at least 12 months. RCD is classified into type 1 RCD (RCD1), which presents a normal intraepithelial lymphocyte phenotype with the surface expression of CD3 and CD8 markers and polyclonal T-cell receptors (TCRs), and into type 2 RCD (RCD2), which is more aggressive with aberrant intraepithelial T-cell clone (IEL), without expression of surface CD3 but with expression of intracytoplasmic CD3. Aberrant IELs frequently present TCR clonality. The diagnosis of RCD1 and RCD2 requires small bowel investigations, such as videocapsule endoscopy and enteroscopy, together with specific IEL analyses (immunohistochemistry, flow cytometry, and molecular biology) [7,8]. MiRNA profiling identified 38 miRNAs distinctive for these CD phenotypes, highlighting that the miR-192/215 family was down-regulated in CD patients despite a GFD [9], but miR-192/215 and miR-200 were consecutively lost in RCD2 and in intestinal T-cell lymphoma (ITL)-CD, whereas miR-19/92 and C19MC miRNAs were up-regulated [10]. In our study, we have extended the research on miRNAs as biomarkers in CD by focusing on the response to a GFD, to improve the stratification and follow-up of patients for the identification of RCD2, which can easily evolve towards malignant pathological phenotypes. Furthermore, to identify their putative targets, we have utilized bioinformatics tools, which are helpful for understanding the impact of miRNAs on gene networks and pathways with the aim to provide a more precise assessment of their functions.

## 2. Materials and Methods

### 2.1. Study Population of Celiac Patients

In our investigations, we considered a total of 16 samples from celiac patients; among them there were 5 patients responsive to a GFD, and 11 refractory patients (*n* = 6 RCD1 and *n* = 5 RCD2). The study was registered on http://clinicaltrials.gov/ (accessed on 4 January 2022) (ref. no. NCT04655495). The diagnosis of RCD1 and RCD2 was made according to internationally adopted criteria and based upon the duodenal T-cell receptor rearrangement (monoclonal in RCD2) and the percentage of aberrant T-cells (CD3s^−^/CD3cyt^+^) at cytofluorimetry [6]. The University of Milan’s Institutional Review Board checked and approved the study protocol in accordance with the Declaration of Helsinki; the Project Identification Code of the Local Ethics Committee’s approval of our study is 89_2019. The protocol was approved by the Ethics Committee of Milan/Area B on 6 February 2019. All the patients gave and signed their informed consent before participation in this study.

We performed two types of comparison with RNA samples from celiac patients. First, we compared RCD patients (random type 1 and 2) with responsive patients to define possible markers of refractory CD, and second, we compared RCD2 with RCD1 samples to identify an association with more severe disease. We have reported the anamnestic and clinical information of the enrolled individuals in Appendix A.

### 2.2. Sample Collection and Extraction of RNA

Total RNA was extracted from serum obtained from celiac patients, previously separated by centrifugation at 3000 rpm for 15 min at 4 °C and stored at −80 °C until extraction. A 300 μL sample of serum was mixed with 1.5 mL of TRIzol^®^ Reagent (Merck Life Science, Milan, Italy); after 3 min at room temperature, 300 μL of chloroform was added to the samples, and the aqueous phase was separated by centrifugation at 12,000× *g* for 15 min at 4 °C. RNA was precipitated with 750 μL of 2-propanol at −80 °C overnight, then recovered by centrifugation at 12,000× *g* for 15 min at 4 °C, washed in 75% ethanol, dried, and resuspended in RNase-free water. RNA was quantified by the NanoDrop spectrophotometer.

### 2.3. miRNA Profiling from Serum of Celiac Patients

A 300 ng aliquot of RNA from each sample was analyzed using human v3 miRNA CSO.12 assays as immunology panels (Diatech Lab Line, Jesi (AN), Italy and nCounter System NanoString Technologies, Seattle, WA, USA). Data were analyzed using nSolver Analysis Software 4.0 (https://www.nanostring.com/products/analysis-solutions/ncounter-analysis-solutions/ (accessed on 4 January 2022)). Before evaluating dysregulated miRNAs, we carried out a quality control analysis to detect any anomalies during the processing of samples and to verify the performance of the reactions. After the background subtraction of the geometric average of all the negative controls, we used the geometric average of all the positive controls as a normalization factor and normalized all miRNAs on the basis of the codeSet content by selecting ‘all gene function’ (excluding Housekeeping, SpikeIn, Ligation). A different miRNA expression was defined using fold change (FC) estimation of the groups, and the *t*-test was performed considering a *p* value < 0.05 with a 95% confidence interval as significant. In addition, we calculated the false discovery rate (FDR) and Bonferroni correction [11].

### 2.4. Data Processing of Targets of miRNAs

The predictive targets of miRNAs were identified using the database miRDB (http://mirdb.org/ (accessed on 4 January 2022)) thanks to the MirTarget bioinformatic tool, which is based on sequencing experiments [12].

### 2.5. Functional Enrichment Analysis and Network Construction

Enrichment and gene ontology (GO) analyses were carried out using the WEB-based Gene SeT AnaLysis Toolkit (WebGestalt) (http://www.webgestalt.org/ (accessed on 4 January 2022)) selecting over-representation analysis (ORA) (as the method for *Homo sapiens*) and gene ontology (as the functional database) [13]. The schemes of miRNAs and their targets, or the target genes and their referred GO, were obtained by Cytoscape_v3.9.0 (https://cytoscape.org/ (accessed on 4 January 2022)) [14]. For protein–protein interaction analysis, we employed NetworkAnalyst (https://www.networkanalyst.ca/ (accessed on 4 January 2022)) [15].

## 3. Results

### 3.1. Details of the Populations Enrolled in the Study

We have investigated miRNA expression in the serum of three cohorts of CD patients composed of responsive to GFD, RCD1, and RCD2. In the first analysis, we compared 5 GFD and 7 RCD, while in the second analysis we compared 5 RCD2 and 6 RCD1 subjects. In Appendix A, we recorded for each patient the following clinical and demographic data: age, age at the diagnosis of CD and RCD, sex, type of RCD, last duodenal histology (Marsh–Oberhuber classification), TG2 antibodies, the presence of aberrant IELs, comorbidities, TCR*γ* rearrangement, ongoing treatment, and compliance with GFD (urine gluten peptide detection). Five CD patients responsive to GFD were analyzed: 4 females, age 42.4 ± 10.43 years, and age at diagnosis of CD 9.2 ± 12.21 years. Eleven RCD patients were analyzed: 7 females, age 54.73 ± 14.79 years, age at diagnosis of CD 46.55 ± 14.40 years, and age at RCD diagnosis 48.55 ± 13.60 years. Six of them were classified as RCD1 and five as RCD2. No differences in terms of sex distribution and age, age at CD, and RCD diagnosis have been evidenced between the two groups. All duodenal histology presented atrophy without differences in terms of the Marsh–Oberhuber classification between RCD1 and 2. Comorbidity was present in at least 33.3% (2/6) of RCD1 patients and 60% (3/5) of RCD2; globally, 45% (5/11) of immunological disorders were present in RCD.

### 3.2. Identification of Marker miRNAs Predictive of Refractoriness in Celiac Patients

We performed the miRNA profiling of serum from CD patients with the aim to identify a panel of miRNAs useful for the screening and monitoring of RCD individuals. In the first analysis, we compared seven RCD patients with five patients responsive to a GFD using Nanostring Technologies and nSolver™ Analysis Software 4.0 to evaluate the serum levels of dysregulated miRNAs. This comparison revealed 264 miRNAs differently expressed with a *p*-value < 0.05. Of these, 141 miRNAs displaying adjusted FDR and Bonferroni correction < 0.05 have been clustered in the heatmap shown in Figure 1 and are listed in Appendix A.

Five miRNAs (miR-27b-3p, miR-206, miR-216b-5p, miR-4536-5p, and miR-548g-3p) were increased with FC > 1.5, whereas the other 136 miRNAs were decreased slightly, although only significantly. Believing that upregulated miRNAs are better suited for diagnostic and prognostic applications, we validated the top five miRNAs in additional samples (2 RCD patients vs. four subjects responsive to GFD) associated from a clinical point of view with the refractory condition. Additional details of the analysis carried out on dysregulated miRNAs in CD patients can be found in Appendix A.

### 3.3. Identification of Marker miRNAs Predictive of Refractoriness to Distinguish RCD1 from RCD2

In the second analysis, we focused on finding possible markers linked to the RCD2 subset. Therefore, we recruited additional patients with RCD for a total of 11 subjects, 5 of whom had RCD2 and 6 RCD1. The search for differentially expressed miRNAs in serum revealed 15 miRNAs with a *p* value < 0.05. Their hierarchical clustering dendrogram is shown in Figure 2, and the FC is reported in Table 1: eight of them were increased (miR-770-5p, miR-181b-2-3p, miR-1193, miR-1226-3p, miR-490-3p, miR-302c-3p, miR-371a-5p, and miR-320e), while seven were decreased in RCD2 (let-7d-5p, miR-606, miR-1306-5p, miR-101-3p, miR-345-5p, miR-935, and miR-107). In addition, we underline that miR-490-3p, let-7d-5p, miR-101-3p, and miR-935 were already dysregulated with respect to the GFD-responsive samples; the levels of miR-490-3p were further raised in RCD2; and the levels of let-7d-5p, miR-101-3p, and miR-935 dropped slightly.

Note that the analysis of the median and the distribution in quartiles plotted in Figure 3 lets us better appreciate the differences in the expression of these miRNAs in relation to the group. Among those showing FC > 2 (as absolute value), that is, miR-770-5p, miR-181b-2-3p, miR-1193, miR-1226-3p, miR-490-3p, and miR-107, the most relevant candidates are miR-770-5p, miR-181b-2-3p, miR-1193, and miR-1226-3p.

### 3.4. Analysis of Targets of miRNAs Dysregulated in RDC2 Patients and Networks of Genes Engaged in Biological Functions

Identified miRNAs in serum can also easily function as modulators of target genes in tissular regions far from their point of release. Hence, the study of their possible targets can be relevant to define networks of transcriptionally modulated genes. Consequently, these targets might play a role in the pathophysiology of alterations occurring in celiac patients, as well as being useful in research on other types of biomarkers. The first goal was to identify targets of miRNAs dysregulated in RCD2. For this purpose, we have used a similar approach to that described by Garcia-Moreno et al. [16], performing an enrichment analysis of the targets of all 15 miRNAs resulting from the findings reported above (Figure 3 and Table 1) using the database miRDB (http://mirdb.org/ (accessed on 4 January 2022)). We considered all the significant miRNAs in order not to lose important information for the subsequent steps to define the affected pathways. We have chosen targets with a score greater than 90%, obtaining a list of 915 genes (some targeted by several miRNAs), reported along with additional information in Appendix A. Further, we carried out an enrichment analysis using WebGestalt to restrict the list to 751 relevant ‘Entrez Gene IDs’, which are annotated within the selected functional categories and in the reference list, as shown by the bar chart of the main macro-categories (Figure 4A).

Focused on biological process not redundant (as category), we set a range of Entrez Gene IDs in the category between 5 and 500 and the Multiple Test Adjustment Benjamini-Hochberg FDR < 0.05 as the significance level. In this manner, we identified 71 enriched categories that we have further filtered to restrict redundancy by choosing weighted set parameters that maximize gene coverage. In this manner, we have obtained 20 GO annotations (reported in Figure 4B and listed in Table 2 based on decreasing enrichment ratio), which describe the major number of relevant genes with respect to those contained in the specific set. There were 397 overlapping genes.

Note that the best categories, taking as the threshold an enrichment ratio > 2, were in response to transforming growth factor β with 32/238 genes of the set; cytoskeleton-dependent intracellular transport with 21/170; the regulation of gene expression with 30/258; in utero embryonic development with 36/345; and, finally, cell–cell signaling by Wnt with 47/460 (see Appendix A). As a miRNA can target multiple genes, and the same gene can also be affected by more than one miRNA, we focused the succeeding investigations into 39 genes targetable by more than one miRNA. In our opinion, these represent the core of possible impacted transcripts. As summarized in Appendix A [17,18,19,20,21,22,23,24,25,26,27,28,29,30,31,32,33,34,35,36,37], we have associated each target gene with putative specific functions, with 29 of them being validated in the literature.

To better understand the relationship between target genes and corresponding miRNAs, we have used Cytoscape_v3.9.0 and schematized the most important relationships in Figure 5. MiR-7d-5p, miR-101-3p, miR-107, and miR-302c-3p were connected with the greatest number of validated targets. Indeed, their relationship was confirmed by miRTarbase and/or Tarbase (DIANA). The following seven validated genes were simultaneously targeted by three miRNAs (usually represented by two downregulated miRNAs and one upregulated miRNA): TNRC6B (trinucleotide repeat-containing gene 6B protein), CPEB3 (cytoplasmic polyadenylation element binding protein 3), MYCN (MYCN proto-oncogene, bHLH transcription factor), FZD6 (frizzled class receptor 6), FBXW7 (F-box and WD repeat domain containing 7), TET3 (Tet methylcytosine dioxygenase 3), and TGFBR3 (transforming growth factor β receptor 3).

Except for the last gene, which is regulated by decreased miRNAs in serum, for the others the effect on expression is more complex because of the opposite activities of miRNAs. Many of the proteins encoded by these target genes are regulatory factors.

For instance, TET3 is engaged by REST to regulate gene expression by hydroxymethylation; CPEB3 is a translational regulator; TNRC6B is a factor playing a role in miRNA- and siRNA-dependent post-transcriptional gene silencing; *MYCN* encodes the well-known proto-oncogene; FBXW7 is a tumor suppressor controlling proteasome-mediated degradation of oncoproteins and is mutated in tumors, including acute myeloid leukemia [38] and 25% of adult T-cell leukemia/lymphoma [39]; and FZD6 belongs to G-protein-coupling receptors playing a role in the non-canonical Wnt. Using Cytoscape_v3.9.0, we demonstrated the relationship between the 39 genes targeted by more than one miRNA and the best 20 GO annotations. As reported in Figure 6, TGFBR3 belongs to the functional process GO:0071559 (the response to transforming growth factor β) along with ACVR2B (the activin receptor type 2B) and the transcription factor ONECUT2 (one cut domain family member 2) involved, respectively, in cell signaling and tumor growth. TNRC6B and MYC are also active in gene-expression regulation [10,40], defined by GO:0040029, SMARCA5 (the SWI/SNF-related, matrix-associated, actin-dependent regulator of chromatin, subfamily A, member 5), and TRIM71 (the tripartite motif containing 71). In this context, SMARCA5 regulates the access to DNA and is upregulated in acute myeloid leukemia [41], while TRIM71 is a post-transcriptional repressor involved in the cell cycle maintaining embryonic stem cell growth and in the processing of the presentation of class I MHC-mediated antigen [42]. CPEB3 is another post-transcriptional regulator that controls the polyadenine tail of mRNA and interferes with different biological functions (GO:0001101, GO:0010608, and GO:0050769) [43]. Again, the functional category GO:0016569 (covalent chromatin modification) includes TET3 and TET2 (Tet methylcytosine dioxygenase 2), with the latter altered in patients with several myeloid malignancies [44,45]; DR1 (the down-regulator of transcription 1), forming the well-known DR1/DRAP1 heterodimer [46]; and ELK4 (ETS transcription factor ELK4), forming ternary complexes with the serum response elements in the promoter of the c-Fos oncogene [47]. Interestingly, FZD6 is involved in GO:0198738 (cell–cell signaling by Wnt) [48] and GO:0002009 (the morphogenesis of an epithelium) [49], functions that may be strongly associated with CD. Finally, FBXW7 and BTG1 (BTG anti-proliferation factor 1) are linked to chromosomal translocation in B-cell chronic lymphocytic leukemia [50,51], and RORA (RAR related orphan receptor A) has an impact on angiogenesis (GO:0001525) and is crucial for the development of type 2 innate lymphoid cells and cytokine production [52,53].

To complete our analysis, we made use of NetworkAnalyst 3.0 to better understand the networks in which the relevant genes are involved. We defined four pathways (Figure 7): (a) the main one is centered on RELA, which is required for NF-κB activation, to which are connected most of the 39 genes targeted by multiple dysregulated miRNAs in the serum of RCD2 patients; (b) the second includes TNRC6B, TRIM71, and AGO4 (protein argonaute-4), which are associated with functions affecting transcript stability; (c) the third includes FZD6 and FZD3, regarding NF-κB-mediated Wnt signaling; and (d) the fourth includes CPEB3 and BTG1, which are relevant for controlling neoplastic proliferation.

### 3.5. Additional Information about the Top miRNAs and GOs

We investigated further target genes of miR-181b-2-3p, miR-1226-3p, and miR-1193 because they are not recognized by multiple miRNAs also excluded by the previous analysis, although they belong to the top five categories (see Table 2). Concerning the genes recognized by miR-181b-2-3p, we found, in the GO:0071559 category (the response to transforming growth factor β), FGFR2 (fibroblast growth factor receptor 2) and nuclear receptor NR3C1 (nuclear receptor subfamily 3 group C member 1) acting as receptors of glucocorticoids; in GO:0198738 (cell–cell signaling by Wnt), we found FGFR2, the X-linked gene DDX3X (DEAD-box helicase 3 X-linked), and FBXW11 (the F-box and WD repeat domain containing 11) associated with lymphocytic leukemia [38,54] and ROR1 (receptor-tyrosine-kinase-like orphan receptor 1), which is essential during embryogenesis and upregulated in tumors; while *CCN1* (cellular communication network factor 1) and the gene coding the new factor VASH1 (vasohibin-1) belong to the GO:0001701 pathway (in utero embryonic development)—they inhibit angiogenesis and are enhanced by VEGF-A and Fibroblast Growth Factor (FGF)-2 [55] (which is a target of the other miR-1226-3p). Considering the genes interacting with miR-1193, one is included in the GO:0071559 category and encodes cysteine protease USP15 (ubiquitin-specific protease 15), which is significantly expressed at low levels in the chronic myelogenous leukemia cell lines and peripheral blood mononuclear cells of affected patients [56]. The other genes were neuronal targets, such as PLPPR5 (phospholipid phosphatase-related 5), which is associated with filopodia and neurite growth and GSG1L (germ cell-specific gene 1-like) [57], a transmembrane auxiliary protein suppressing current flow acting on neuronal calcium-permeable AMPA receptors responsible for excitatory synaptic transmission in the brain. Their putative involvement in this pathological context obviously needs further investigation.

## 4. Discussion

Celiac disease (CD) usually has a favorable prognosis and a good response to a gluten-free diet (GFD), but in a limited number of patients, severe complications can occur if they are unresponsive to a GFD. Previous results indicated the deregulation of seven miRNAs (miR-31-5p, miR-192-3p, miR-194-5p, miR-551a, miR-551b-5p, miR-638, and miR-1290) in a larger series of CD patients with specific clinical phenotypes [9]. These seven miRNAs were studied in duodenal mesenchymal cells obtained from CD patients and treated with gliadin peptides (13- and 33-mer). Again, the miRNA cluster miR-192/194, involved in matrix remodeling, was deregulated in CD with different clinical presentations, and the miR-192-3p levels were modulated by gliadin peptides in vitro. In refractory CD (RCD), small bowel malignancies, enteropathy-associated T-cell lymphoma (EATL), or non-Hodgkin lymphoma are serious complications with a high mortality rate. RCD can be further classified as RCD1 or RCD2 depending on the IEL phenotype. Actually, RCD2 is considered a preneoplastic condition and treated with chemotherapy or immunosuppressive agents, although with a low response rate. Furthermore, the absence of biomarkers indicating the patients at a high risk of developing malignancies reduces the efficacy of diagnostic interventions. Previous studies indicated that miRNAs could play a relevant role as diagnostic biomarkers and/or therapeutic targets [58,59]. For example, the miR-200 and miR-192/215 families were progressively undetectable in RCD2 and lymphomas, while miR-17/92 and C19MC were up-regulated. These miRNAs are decreased in IL15-transgenic mice after Janus kinase (JAK) inhibition. According to miRNA expression, SMAD3, MDM2, c-Myc, and activated-STAT3 were inversely regulated and increased in RCD2 and EATL, after JAK inhibition restoring their baseline levels. These findings suggested the miRNA-related activation of STAT3 and c-Myc oncogenic signaling in RCD2, contributing to lymphomagenesis [10].

We have performed the present study employing 7 versus 5 and 6 vs. 5 subjects to compare different groups of RCD individuals against GFD and RCD2 against RCD1 respectively. The size of the samples is not completely respective of the indications found in the literature [60], suggesting at least 6 vs. 6 biological replicates, but, in any case, they are indicative and close to them, considering the rarity of these clinical conditions. For these reasons, we also applied FDR and Bonferroni correction < 0.05. Firstly, we identified a set of 141 circulating miRNAs deregulated in RCD compared with subjects responsive to a GFD, and 15 miRNAs specifically deregulated in RCD2 compared with RCD1. Among the latter set, eight miRNAs showed increased levels (miR-770-5p, miR-181b-2-3p, miR-1193, miR-1226-3p, miR-490-3p, miR-302c-3p, miR-371a-5p, and miR-320e), while the other seven were decreased in RCD2 (let-7d-5p, miR-606, miR-1306-5p, miR-101-3p, miR-345-5p, miR-935, and miR-107). Some of them (miR-490-3p, let-7d-5p, miR-101-3p, and miR-935) were already differentially expressed in the comparison between RCD and GFD-responsive samples, but note that the levels of miR-490-3p (targeting the KDM7A histone demethylase) were further raised in RCD2, while the levels of let-7d-5p, miR-101-3p, and miR-935 dropped slightly. Focusing on those increasing selectively in RCD2, we point to miR-770-5p, miR-181b-2-3p, miR-1193, and miR-1226-3p as the most relevant candidates. Notably, miR-181b is strongly associated with inflammation, but only a little information has been reported in the literature on the specific miR-181b-2-3p, which seems more appropriately linked to drug response/resistance than to doxorubicin in cancer [61,62]. miR-1193 inhibits fibroblast proliferation, is associated with the apoptosis of fibroblasts exposed to interleukin-1β in a rheumatoid arthritis model [63], and plays a role as an oncosuppressor in human T-cell leukemia cells [64]; monitoring changes in their levels could be relevant as it may possibly be related to the appearance of malignant pathologies. In contrast, functionally miR-770-5p and miR-1226-3p promote tumorigenicity in several types of cancers [65,66]. We further carried out our investigations using bioinformatic tools to define the possible networks of genes engaged in the category ‘biological process’ and the target of the regulated set of miRNAs in RCD2. We obtained 20 GO annotations containing the main relevant genes (*n* = 397), among which the top five GO were ‘response to transforming growth factor β’ (including 32/238 genes of the set); ‘cytoskeleton-dependent intracellular transport’ (21/170); ‘the regulation of gene expression’ (30/258); ‘in utero embryonic development’ (36/345); and, finally, ‘cell–cell signaling by Wnt’ (47/460). Among the targets of the 15 dysregulated miRNAs in RCD2, we focused on those that have more than one target gene and are validated in the literature. Several of the evidenced genes are notoriously linked with CD, belonging to the ‘response to Transforming Growth Factor β’ pathway [1,2], or ‘cell–cell signaling by Wnt’ [67] and ‘the regulation of gene expression, epigenetic’ [68] or interconnected in the network of RELA. Concerning the putative targets of the best miRNA candidates, the genes recognized by miR-181b-2-3p are involved in mitotic processes (CEP135) [43] or bind RNA and are active in P-body and nucleoplasm (TNRC6B) [40] and angiogenesis (DR1) [46], as well as being the target of miR-1226-3 (the oncosuppressor FBXW7) [38,39,50], while miR-1193 specifically affects the genes of the TGF-β pathways. We want to also highlight the ‘covalent chromatin modification’ network, containing TET2 [37,44,45], which is reversibly modulated by miR-302c-3p and miR-101-3p, because it is a potential oncogene and is affected by mutations associated with EATL in RCD2 patients [69]. From this perspective, the miRNAs related to specific genetic features interfering with chromatin structure and affecting transcriptional regulation assume increasing importance. Analogously, we underline as crucial another regulatory network of genes, including AGO2 [21,28], AGO4 (associated with downregulation of miR-107 and miR-let-7d-5p), and DICER1, belonging to the machinery associated with post-transcriptional control.

For a clearer understanding, we have summarized these conclusions in the scheme below (Table 3). With this study, we highlight that to understand the shift from RCD1 to RDC2, it is not enough to consider the wide range of genes related to inflammation; alterations and the dysregulation of the indicated genes (oncogenes, tumour suppressors, regulators) responsible of reprogramming and promoting changes through pre- and post-transcriptional mechanisms need to be examined as well.

## 5. Conclusions

We have identified a set of miRNAs characterizing the RCD2 trait that can be monitored in the serum of patients. Some of them are associated with an increase in inflammation, while others signal the progression of pathology as they are closely linked to oncogenesis. Interestingly, some deregulated miRNAs could reflect altered conditions derived from epigenetic or post-transcription modifications that could be better investigated by whole-genome analysis.

## Figures and Tables

**Figure 1 biomedicines-10-01408-f001:**
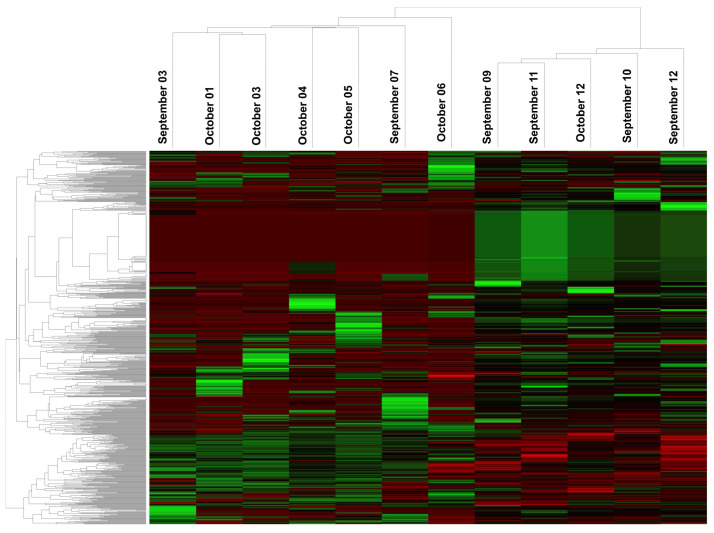
An expression heatmap of deregulated miRNAs in unresponsive patients vs. patients responsive to a GFD. The upregulated miRNAs are indicated in Red, and the downregulated miRNAs are indicated in Green. *p* value < 0.05, and FDR and Bonferroni correction < 0.05.

**Figure 2 biomedicines-10-01408-f002:**
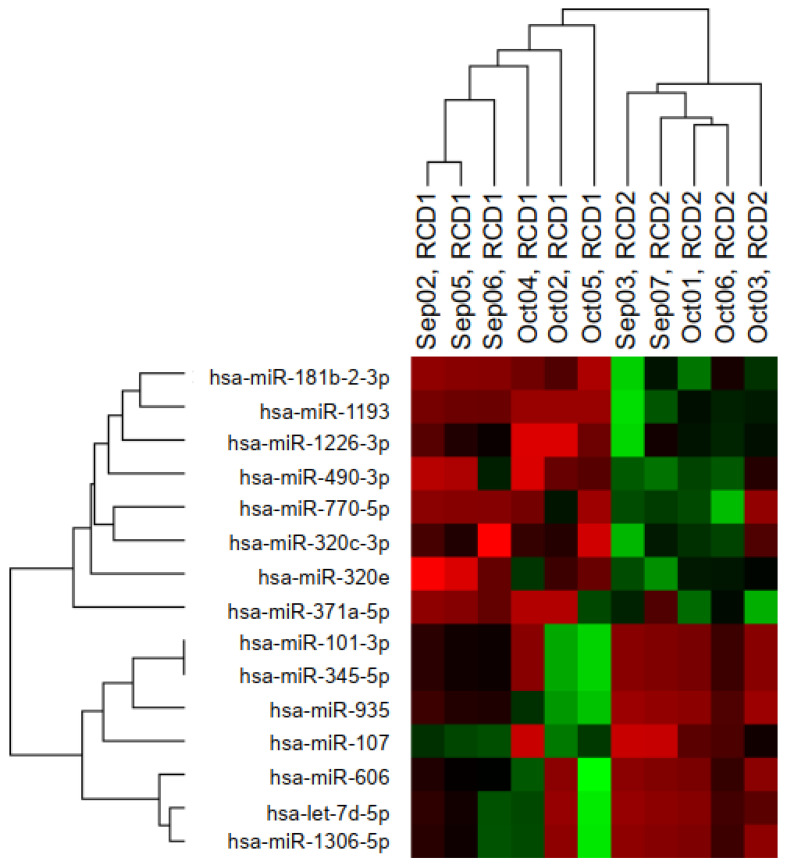
An expression heatmap of the deregulated miRNAs in RCD2 (*n* = 5) vs. RCD1 (*n* = 6) patients. The upregulated miRNAs are indicated in Red, and the downregulated miRNAs are indicated in Green. *p* value < 0.05.

**Figure 3 biomedicines-10-01408-f003:**
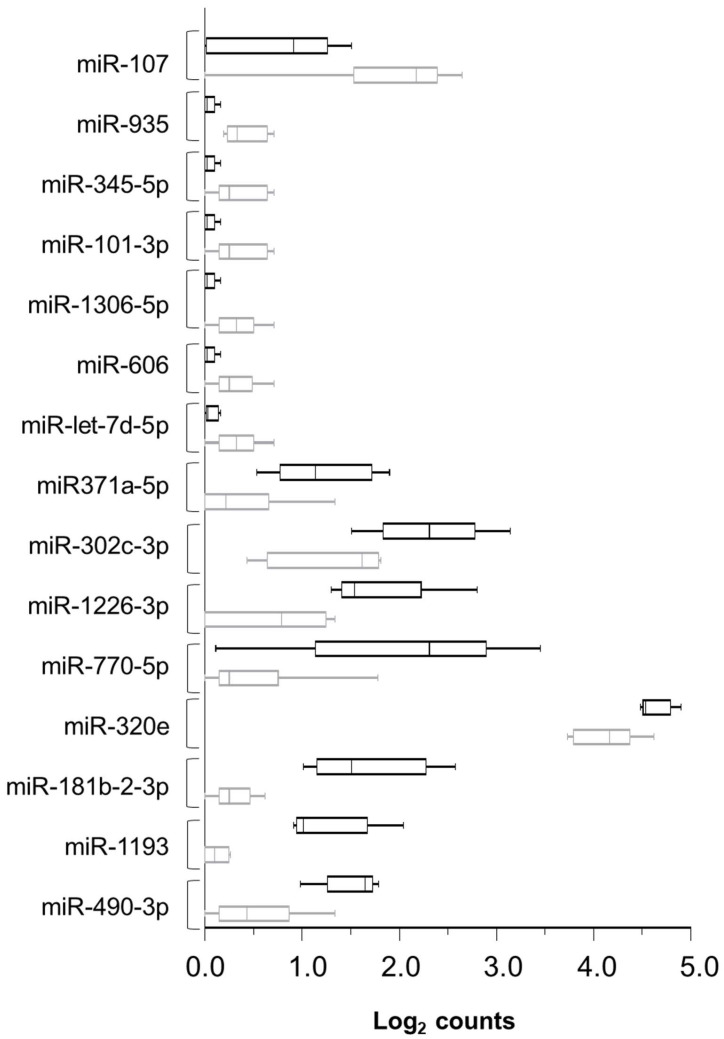
The expression of dysregulated miRNAs in serum from RCD2 and RCD1 patients. The box plot showed the levels of expression of 15 miRNAs analyzed by Nanostring Technologies and by nSolver Analysis Software 4.0. The black boxes indicate samples from RCD2 subjects, and the grey boxes, the samples from RCD1 patients. The median and quartile distribution are plotted.

**Figure 4 biomedicines-10-01408-f004:**
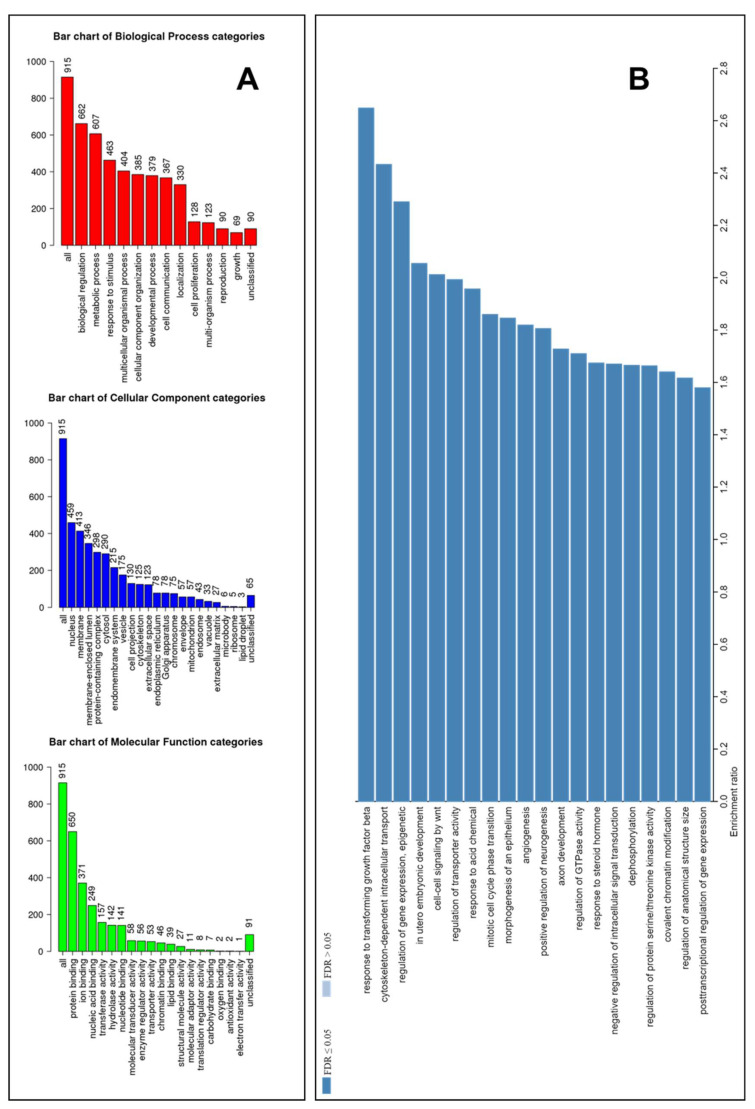
An enrichment analysis by the WEB-based Gene SeT AnaLysis Toolkit (WebGestalt) and main gene ontology (GO) annotations of the biological process category. (**A**) A GO Slim summary of the uploaded Entrez Gene IDs. The following categories were represented: biological process in red; cellular component in blue, and molecular function in green. The number of IDs present in the category are listed on top of the bar. (**B**) The GO annotations resulting from enrichment and maximizing gene coverage of the non-redundant biological process category displayed with respect to the enrichment ratio.

**Figure 5 biomedicines-10-01408-f005:**
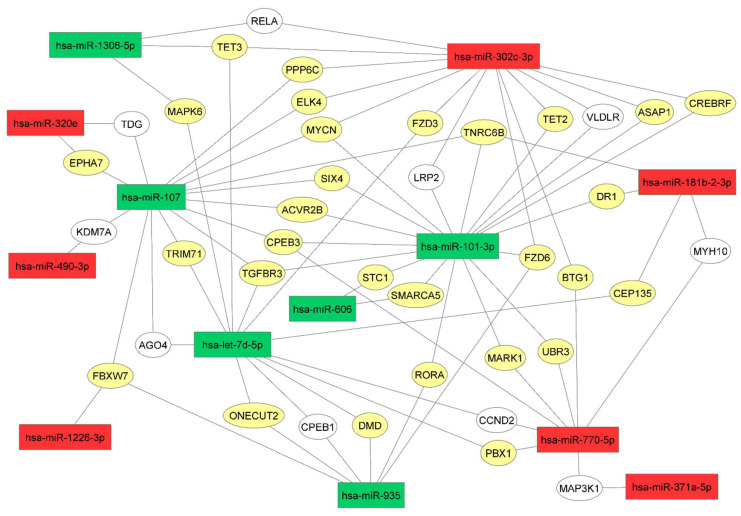
The network of genes targeted by more than one miRNA, characterizing RCD2. An analysis performed using the tool Cytoscape_v3.9.0. The miRNAs upregulated in the serum of patients are indicated in red, and the downregulated miRNAs are indicated in green. The targets genes of miRNAs reported in the literature and validated using other bioinformatic tools, miRTarbase and/or Tarbase (DIANA), are indicated in yellow, while the predicted target previously analyzed by the database miRDB is indicated in white.

**Figure 6 biomedicines-10-01408-f006:**
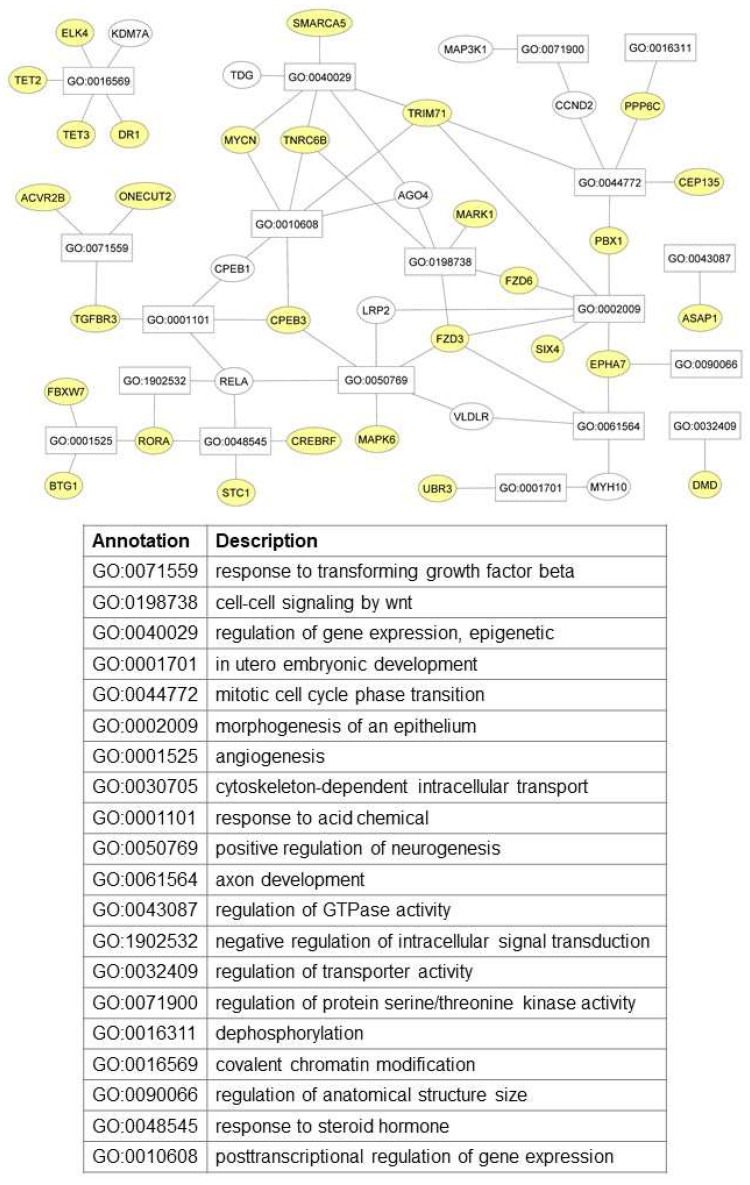
The network describing the relationships between the genes targeted by more than one miRNA and the gene ontology (GO) associated with biological processes. The analysis was performed using the tool Cytoscape_v3.9.0. The network is reported at the top (the validated targets are shown in yellow), while the GO annotations are shown at the bottom.

**Figure 7 biomedicines-10-01408-f007:**
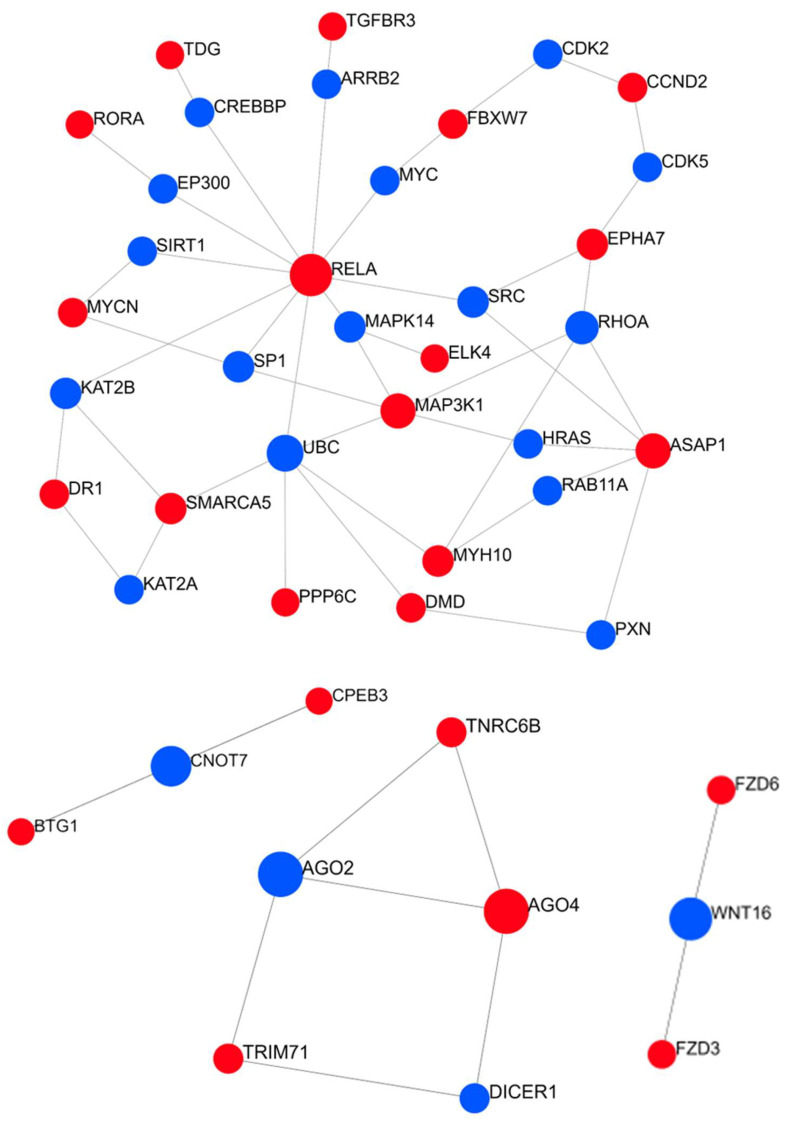
The pathways of protein–protein interactions among the targets of miRNAs dysregulated in the serum of RCD2 patients. The network was generated using NetworkAnalyst 3.0. The target genes representing our input are shown in red, while the remaining genes present in the database are shown in blue.

**Table 1 biomedicines-10-01408-t001:** The dysregulated miRNAs evaluated in serum samples obtained from RCD2 and RCD1 patients.

miRNA Name	Fold Change, RCD2 vs. RCD1	*p* Value
hsa-miR-770-5p	3.02	0.04
hsa-miR-181b-2-3p	2.62	0.01
hsa-miR-1193	2.19	0.00
hsa-miR-1226-3p	2.10	0.02
hsa-miR-490-3p	2.01	0.00
hsa-miR-302c-3p	1.98	0.02
hsa-miR-371a-5p	1.81	0.02
hsa-miR-320e	1.41	0.01
hsa-let-7d-5p	−1.20	0.04
hsa-miR-606	−1.20	0.04
hsa-miR-1306-5p	−1.22	0.03
hsa-miR-101-3p	−1.23	0.04
hsa-miR-345-5p	−1.23	0.04
hsa-miR-935	−1.29	0.01
hsa-miR-107	−2.29	0.04

**Table 2 biomedicines-10-01408-t002:** The gene ontology (GO) annotations derived from genes targeted by miRNAs detected in serum from RCD2 patients. The analysis was post-processed using a weighted set cover method to reduce redundancy. The GO are listed with respect to the enrichment ratio.

Annotation	Description	Genes in the Set	Expected Value	Enrichment Ratio	*p* Value < 0.05	FDR < 0.05
GO:0071559	Response to transforming growth factor β	32/238	12.070	2.6513	0.00438	0.00037
GO:0030705	Cytoskeleton-dependent intracellular transport	21/170	86.211	2.4359	0.00014	0.00809
GO:0040029	Regulation of gene expression, epigenetic	31/258	13.084	2.2929	0.00002	0.00399
GO:0001701	In utero embryonic development	36/345	17.496	2.0576	0.00003	0.00399
GO:0198738	Cell–cell signaling by Wnt	47/60	23.328	2.0148	0.00000	0.00158
GO:0032409	Regulation of transporter activity	25/247	12.526	1.9959	0.00079	0.02165
GO:0001101	Response to acid chemical	33/332	16.837	1.9600	0.00017	0.00885
GO:0044772	Mitotic cell cycle phase transition	46/487	24.697	1.8626	0.00004	0.00399
GO:0002009	Morphogenesis of an epithelium	45/480	24.342	1.8487	0.00005	0.00439
GO:0001525	Angiogenesis	45/487	24.697	1.8221	0.00007	0.00562
GO:0050769	Positive regulation of neurogenesis	41/447	22.668	1.8087	0.00018	0.00885
GO:0061564	Axon development	43/490	24.849	1.7304	0.00032	0.01315
GO:0043087	Regulation of GTPase activity	41/472	23.936	1.7129	0.00055	0.01856
GO:0048545	Response to steroid hormone	33/388	19.676	1.6771	0.00257	0.03705
GO:1902532	Negative regulation of intracellular signal transduction	42/495	25.103	1.6731	0.00075	0.02137
GO:0016311	Dephosphorylation	39/461	23.378	1.6682	0.00121	0.02707
GO:0071900	Regulation of protein serine/threonine kinase activity	42/497	25.204	1.6664	0.00082	0.02170
GO:0016569	Covalent chromatin modification	39/468	23.733	1.6433	0.00160	0.02954
GO:0090066	Regulation of anatomical structure size	40/487	24.697	1.6196	0.00183	0.03028
GO:0010608	Post-transcriptional regulation of gene expression	39/486	24.646	1.5824	0.00311	0.04135

**Table 3 biomedicines-10-01408-t003:** A scheme of the relations between miRNAs and targets genes with an impact on cancer. Here are reported the only validated genes targeted by more than one miRNA. (*) Already present in GFD-responsive. In Red, the best candidates as markers of RCD2 trait. Up-arrows indicate an increase and down-arrows indicate a decrease of the miRNA in RCD2.

miR	Target	Function	Specific Role
↓ miR-101-3p *	CPEB3	Translational repressor	Represses the transcription of the STAT5B target gene EGFR
↑ miR-770-5p			
↓ miR-107			
↓ miR-935 *	FBXW7	Phosphorylation-dependent ubiquitination	Colorectal cancer and ovarian serous cystadenocarcinoma, which are involved in the NOTCH signaling
↑ miR-1226-3p			
↓ miR-107			
↓ miR-935 *	FZD6	Negative regulator	The inhibition of the canonical Wnt/beta-catenin signaling cascade
↓ miR-101-3p *			
↑ miR-302c-3p			
↓ miR-101-3p *	MYCN	Proto-oncogene, DNA-binding transcription factor	Involved in the apoptosis, autophagy, and NOTCH pathways
↑ miR-302c-3p			
↓ miR-107			
↓ let-7d-5p *	TET3	Methylcytosine dioxygenase	Plays a role in the DNA methylation process
↑ miR-302c-3p			
↓ miR-1306-5p			
↓ let-7d-5p *	TGFBR3	Binds to TGF-β	Inhibits TGFB signaling
↓ miR-101-3p *			
↓ miR-107			
↓ miR-101-3p *	TNRC6B	Mediated gene silencing by miRNAs and siRNAs	RET signaling, PI3K/AKT activation
↑ miR-181b-2-3p			
↓ miR-107			
↓ miR-101-3p *	ACVR2B	Transforming growth factor-beta family	IGF1-Akt signaling
↓ miR-107			
↓ let-7d-5p*	AGO4	Mediated gene silencing by miRNAs	Post-transcriptional control
↓ miR-107			
↑ miR-302c-3p	BTG1	Putative tumor suppressor	B cell lymphocytic leukemia
↑ miR-770-5p			
↓ let-7d-5p *	CEP135	Centrosomal protein	The regulation of PLK1 activity at G2/M transition
↑ miR-181b-2-3p			
↓ miR-101-3p *	DR1	DR1/DRAP1 heterodimer with TBP, repressor	Represses class II genes, chromatin regulation/acetylation
↑ miR-181b-2-3p			
↑ miR-302c-3p	ELK4	Transcriptional factor	Binding c-fos proto-oncogene promoter, ERK signaling
↓ miR-107			
↓ let-7d-5p *	FZD3	Phosphorylation-dependent ubiquitination	The inhibition of GSK-3 kinase, the nuclear accumulation of β-catenin, and the activation of Wnt target genes
↑ miR-302c-3p			
↓ miR-935 *	ONECUT2	Transcription factors	Stimulates expression
↓ let-7d-5p *			
↑ miR-302c-3p	RELA	Multi-ligand endocytic receptor	Critical for the reuptake of numerous ligands, (lipoproteins, sterols, vitamin-binding proteins, and hormones)
↓ miR-1306-5p			
↓ miR-935 *	RORA	Orphan receptor	A regulator of embryonic development, cellular differentiation, and cytokine signaling in the immune system
↓ miR-101-3p *			
↓ miR-101-3p	SMARCA5	Actin-dependent regulator of chromatin	Regulates the access to DNA and is upregulated in acute myeloid leukemia
↓ miR-606			
↓ miR-101-3p *	TET2	Methylcytosine dioxygenase	Involved in myelopoiesis
↓ let-7d-5p *	TRIM71	E3 ubiquitin-protein ligase	Binds miRNAs, which are involved in the G1-S phase of the cell cycle of embryonic stem cells; prevents premature differentiation, Class I MHC-mediated antigen processing, and the presentation pathway

## Data Availability

Not applicable.

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
