# Peer review of "Circulating microRNAs Suggest Networks Associated with Biological Functions in Aggressive Refractory Type 2 Celiac Disease"

_biomedicines, 2022, doi:10.3390/biomedicines10061408_

Round 1

Reviewer 1 Report

To date, only effective treatment for celiac disease is a lifelong gluten-free diet (GFD); there are no markers that allow the identification of various stages of the disease, nor potential therapeutic targets. The pathological state becomes even more critical in the case of the onset of Refractory celiac disease (RCD), in particular  RCD type 2 (RCD2).

The RCD  is characterized  by a lack of response to a gluten-free diet after six to 12 months of treatment  in  the 5% of  CD patients. RCD presents the symptoms persistence, intestinal damage and an abnormal population of T cells in the gut,  is also accompanied by severe anemia. In the worst clinical course the RCD type 2 could evolve into intestinal lymphoma, and also in this case,  the absence of biomarkers to identify patients at high risk of developing malignancies ,  does not allow any type of effective intervention.

In this study, the final goal of the authors is identify, in serum of CD patient in Gluten Free Diet, miRNAs usefull as biomarkers to discriminate subjects in remission from the others one not responsive to GFD , to improving pathological stratification and follow-up. With particular attention to the identification of mRNAs able to discriminate between type 1 RCD and type 2 RCD. the latter can easily evolve towards pathological malignant phenotypes

The authors conduct  two types of analysis; first  the comparison  between  RCD patients (belonging to both groups 1 and 2) with respect to the patients responsive to GFD in order to identify miRNAs markers of refractory CD.  Second, compare subjects RCD2 respect RCD1 to identify miRNAs markers associated with a more severe course of the disease.

After RNA extraction from sera of the subjects included in the study,  the authors performed a MiRNA profiling, identifying  dysregulated miRNAs and evaluating  statistically the confidence of their fold change.

Finally, using bioinformatics tools,  targets of selected miRNAs have been identified, in order to understand their impact on gene networks underlying the pathology.

The result interstingly shows the identification of a  set of miRNAs  dysregulated  mainly associated to inflammatory profiles , some related to conditions derived from epigenetic or post-transcriptional alterations. Even more relevant is the finding that the dysregulated miRNA in the RCD2 trait  are closely linked to oncogenesis.

Question and suggestion:

1)      Most cases of refractory celiac disease occur in older patients who are not diagnosed until later in life. It would be possible to add a table with the information of the enrolled patients, for example age, sex, other pathologies, therapies, etc. in my opinion it is important information necessary to define the comparability of the individuals included in the study.

2)      For the prediction of the miRNAs dysregulated targets, a single bioinformatics tool was used, or the results were be validate with another one?

3)      Some figures are partially visible, for example figure 4A, histograms are missing. it is necessary to improve the quality of all figures. The same for the tables, they are too big.

4)      It is useful to insert a scheme that summarizes the dysregulated miRNAs and their trend between controls and RCD, and between RCD1 and RCD2, and finally the putative target.

Author Response

Reply to Reviewer 1

Point 1: Most cases of refractory celiac disease occur in older patients who are not diagnosed until later in life. It would be possible to add a table with the information of the enrolled patients, for example age, sex, other pathologies, therapies, etc. in my opinion it is important information necessary to define the comparability of the individuals included in the study.

We have included a paragraph in Results with description of patients as summarized in the supplementary Table S1 (in Materials and Methods), with the following information: Sex, Age, Age at CD diagnosis, Age at RCD diagnosis, Duodenal histology, TG2 antibodies, Aberrant IEL (%), Comorbidities, TCRγ, Therapy, GIP results. As you can see RCD patients are old, roughly in the same age of GFD patients, who however were usually identified in the youth.

Point 2: For the prediction of the miRNAs dysregulated targets, a single bioinformatics tool was used, or the results were be validate with another one?

We specified “…in Figure 5. MiR-7d-5p, miR-101-3p, miR-107 and miR-302c-3p were connected with the greatest number of validated targets. Indeed, their relationship was confirmed by miRTarbase and/or Tarbase (DIANA)”, as well as in the legend of Figure 5: “….. In Yellow, targets genes of miRNAs reported in the literature and validated using other bioinformatic tools, miRTarbase and/or Tarbase (DIANA), while in White are indicated the predicted target previously analyzed by the database miRDB.”

Point 3: Some figures are partially visible, for example figure 4A, histograms are missing. it is necessary to improve the quality of all figures. The same for the tables, they are too big.

We apologize for the format of the figures, we have tried to improve the Figure 4, as well as the others. In addition, we have moved a really long table (ex-Table 3) to the supplementary files (now Table S2) and changes the Supplementary material 1 to Table S1. Finally, we have reduced the characters of the other tables.

Point 4: It is useful to insert a scheme that summarizes the dysregulated miRNAs and their trend between controls and RCD, and between RCD1 and RCD2, and finally the putative target.

We summarized the conclusions in Table 4 at the end of the Discussion, focusing on dysregulated miRNAs in RCD2 patients in relation to relevant cancer-associated functions: “…. For a clearer understanding we have summarized these conclusions in the scheme below (Table 3). With this study we highlight that to understand the shift from RCD1 to RDC2 it is not enough to consider the wide range of genes related to inflammation, but it should look at alterations and dysregulation of the indicated genes (oncogenes, tumour suppressors, regulators) responsible of reprogramming and promoting changes through pre- and post-transcriptional mechanisms”

Finally, we have added some references.

Reviewer 2 Report

Suggest separate section 2.1 into study population and include the first 4 lines of section 2.2 on the comparison groups

2.2 Sample collection and RNA extraction

2.3 microRNA profiling

2.4 Data processing and analysis

2.5 Functional enrichment analysis and network construction

Similarly the organization of the results, should start with characteristics of the study population in a table, then discuss each comparison in separate sections.

Fig. 5 is trimmed. Legend for white and yellow nodes should be described.

Fig. 7 is trimmed.

In discussion, prior literature investigating significantly mentioned GO terms in the results should be added.

Recommend adding limitation of sample size.

Adding clear message/recommendation at the end of discussion/conclusion.

Author Response

Reply to Reviewer 2

We thank the Reviewer 2, whose comments helped to better outline this work. We apologize for the bad setting of the figures, of which we also attach the originals.

We believe we have followed all his instructions, detailed below.

Presentation of Materials and Methods (section 2) has been reorganized in 5 sections 2.1 to 2.5 as suggested and the Results have been modified adding a specific section 3.1 to define in detail the structure of the population and we have splitted in 3.2 and 3.3 the analysis RCD versus GFD and RCD2 versus RCD1.

About Figure 5, we have modified the figure and clarified its legend: “… In Yellow, targets genes of miRNAs reported in the literature and validated using other bioinformatic tools, miRTarbase and/or Tarbase (DIANA), while in White are indicated the predicted target previously analyzed by the database miRDB.”

About Figure 7, we have modified the figure.

We have included new references in results and discussion.

In addition, in Discussion we have specified the limitation of sample size: “….. We have performed the present study employing 7 versus 5 and 6 versus 5 subjects to compare different groups of RCD individuals against GFD and RCD2 against RCD1 respectively. The size of the samples is not completely respective of the indications found in the literature [48], suggesting at least 6 versus 6 biological replicates, but are in any case indicative and close to them, considering the rarity of this clinical conditions. For these reasons we applied also FDR and Bonferroni correction < 0.05.”

Finally, we have added in Discussion a scheme (Table 4) for a clearer understanding: “….For a clearer understanding we have summarized these conclusions in the scheme below (Table 3). With this study we highlight that to understand the shift from RCD1 to RDC2 it is not enough to consider the wide range of genes related to inflammation, but it should look at alterations and dysregulation of the indicated genes (oncogenes, tumour suppressors, regulators) responsible of reprogramming and promoting changes through pre- and post-transcriptional mechanisms.”

Finally, we have added some references and improved English.

Round 2

Reviewer 2 Report

Fig 2 need assignment of group in the x axis.